# Beyond NPK: Mineral Nutrient-Mediated Modulation in Orchestrating Flowering Time

**DOI:** 10.3390/plants12183299

**Published:** 2023-09-18

**Authors:** Sang Eun Jun, Jae Sun Shim, Hee Jin Park

**Affiliations:** 1Department of Molecular Genetics, Dong-A University, Busan 49315, Republic of Korea; junse@dau.ac.kr; 2School of Biological Science and Technology, College of Natural Sciences, Chonnam National University, Gwangju 61186, Republic of Korea; 3Department of Biological Sciences and Research Center of Ecomimetics, College of Natural Sciences, Chonnam National University, Gwangju 61186, Republic of Korea

**Keywords:** flowering time, reproductive stage, secondary major nutrient, micronutrients, cofactor, transcription factor, gene expression

## Abstract

Flowering time in plants is a complex process regulated by environmental conditions such as photoperiod and temperature, as well as nutrient conditions. While the impact of major nutrients like nitrogen, phosphorus, and potassium on flowering time has been well recognized, the significance of micronutrient imbalances and their deficiencies should not be neglected because they affect the floral transition from the vegetative stage to the reproductive stage. The secondary major nutrients such as calcium, magnesium, and sulfur participate in various aspects of flowering. Micronutrients such as boron, zinc, iron, and copper play crucial roles in enzymatic reactions and hormone biosynthesis, affecting flower development and reproduction as well. The current review comprehensively explores the interplay between microelements and flowering time, and summarizes the underlying mechanism in plants. Consequently, a better understanding of the interplay between microelements and flowering time will provide clues to reveal the roles of microelements in regulating flowering time and to improve crop reproduction in plant industries.

## 1. Introduction

Flowering time is one of the important agricultural traits alongside germination rate, plant height, yield, drought tolerance, and nutrient use efficiency (NUE). The uniformity and time of flowering, and the transition from vegetative to reproductive stage are influenced by its surrounding environmental conditions such as daylength (photoperiodic flowering pathway), phytohormones such as gibberellic acid (GA) (hormonal flowering pathway), temperature, especially long-period cold temperature (wintertime) (vernalization), plant age, and endogenous signals (autonomous pathway). The molecular mechanisms for each pathway have been characterized for the past two decades [1,2].

Abiotic stress is the adverse impact of non-living factors on plants and their surrounding environment such as drought (dehydration), high and low (or freezing) temperatures, salt, and nutrient deficiency. The type of abiotic stress, duration of the unfavorable condition, and/or recombination of more than one stress condition influence the potential of the plant to perceive input signal(s) and respond to the stress. Environmental stress, especially during the reproductive stage, induces pollen sterility, pollen tube deformation, ovule abortion, and changes in flowering time, which eventually cause decreased yield and crop quality [1,3,4,5,6,7]. Fertilizers and plant nutrient solutions are applied as a common practice to alleviate abiotic stress and enhance crop yield [3,8,9]. Recently, nutrient deficiency (references in [10,11,12]), imbalance of nutrients [13,14,15,16], and heavy metal accumulation in soil [17,18] have been reported to influence flowering time [8], leading to altered plant growth.

The deficiency or imbalance of three major and primary nutrients, nitrogen (N), phosphorus (P), and potassium (K), results in altered flowering time in plants [10]. Previously, NPK fertilizers have been used to control flowering time in crop plants [19,20]. A recent study documented the process of how major nutrient elements influence flowering time and the genes involved in NPK nutrient-mediated regulation of flowering time [10]. Largely, loss of function of genes encoding nutrient transporter proteins localized in the plasma membrane or subcellular organelles and signaling genes involved in nutrient assimilation, consequently responsible for nutrient (N, P, and K) deficiency, are reported to induce flowering [10]. The deficiency or excessive accumulation (or supply) of nutrient elements in the soil might directly affect changes in flowering timing. However, the intricate network involving all elements participating in regulating flowering time in plants has not yet been explored.

This study focuses on how secondary nutrients (calcium, magnesium, and sulfur) and micronutrients (boron, zinc, iron, and copper) influence the flowering process in plants. In addition, we summarize the genes and pathways involved in the nutrient element-mediated regulation of the switch to flower or flowering time in various plants.

## 2. Secondary Nutrients and Flowering Time

### 2.1. Calcium (Ca)

Calcium is critical for plant growth and development, hormonal response, and environmental stress adaptation [21]. Ca^2+^ is also a ubiquitous second messenger in living organisms [22]. Cytosolic calcium ion (Ca^2+^) oscillates in a 24-h rhythmic pattern, and the amplitude and kinetics of calcium fluctuations, the calcium signature, reflect the day-length (photoperiod) and light intensity [23,24]. The circadian clock regulates calcium levels in the cytosolic compartment and integrates oscillator time with environmental signals [23]. The intracellular Ca^2+^ interacts with (i) calmodulin (CaM) and CaM-like (CML) proteins; (ii) calcineurin-B-like (CBL) proteins; and (iii) Ca^2+^—dependent protein kinases (CDPKs or CPKs). These proteins perceive transient changes in calcium levels upon various environmental stresses. Further, they transduce and regulate stress response by interacting with a broad range of target proteins, including membrane transporters (channels, pumps, and ion antiporters), transcription factors, protein kinases, protein phosphatases, and metabolic enzymes [25,26,27,28,29].

The research on how calcium influences flowering began in the late 1980s by applying calcium ions (CaCl_2_) and calcium ionophores, calcium blockers (La^3+^, LaCl_3_), calmodulin inhibitors (chlorpromazine and W7), and calcium chelator (EGTA) in *Pharbitis nil*, which is a short-day (SD, shorter daylength than nightlength) flowering plant and exhibits photoperiodic floral induction at the early seedling stage [30,31]. Since calcium ions and CaM proteins were considered involved in flower induction in *P. nil* [32], functions of CML proteins in other plants related to flowering time have also been characterized. The Arabidopsis deficient mutants of *CML24* or both *CML23* and *CML24* are insensitive to ABA and tolerant to CoCl_2_, molybdic acid (Na_2_MoO_4_), and ZnSO_4_. Interestingly, the mutants exhibit delayed flowering in long-day (LD, longer daylength than nightlength) conditions with higher expression of *FLOWERING LOCUS C* (*FLC*), a floral repressor in LD [33,34]. Recently, CML24, known to regulate circadian oscillation of cytosolic Ca^2+^, was characterized to decipher the information about photoperiod, timing, and light intensity [35]. However, whether CML24 directly regulates photoperiodic flowering or flowering time indirectly by regulating the circadian oscillator(s), which senses the changes in light periods and influences flowering, needs further clarification. 

In *P. nil*, CML was shown to be a critical flowering regulator. However, PnCDPK1 potentially controls the transition of a vegetative shoot apex to a flower bud, as kinases can transduce calcium signals via reversible phosphorylation of target proteins [36,37]. CDPKs and calcium/calmodulin-binding kinases are involved in flowering time in other plants (Arabidopsis, soybean, and tobacco). The knockout mutant of *AtCPK32* exhibits late flowering by increasing the *FLC* expression. CPK32 was found to interact with and phosphorylate FLOWERING CONTROL LOCUS A (FCA) in a Ca^2+^-dependent manner, which induces distal polyadenylation of FCA and reduces the *FLC* expression [38]. A mutation in *CDPK38* (or *CPK38*) developed using CRISPR techniques showed late flowering with lower expression of *FLOWERING LOCUS T* (*FT*) and flowering-related genes in soybean [39]. Tobacco Ca^2+^/calmodulin-binding protein kinase 1 (CBK1) is a negative regulator of flowering. CBK1 binds to calmodulin in a Ca^2+^-dependent manner and phosphorylates itself and its substrates. Transgenic tobacco plants overexpressing *NtCBK1* display a late-flowering phenotype, but not through influencing either the expression of neurofilament light chain (NFL), a tobacco homolog of LEAFY (LFY) affecting floral structure, or by blocking demethylation, a treatment that causes early flowering in tobacco [40]. AtCBK3, also known as AtCRK1 (CDPK-related protein kinase 1), binds to calmodulin (CaM) in a Ca^2+^-dependent manner, while autophosphorylation and substrate phosphorylation activities are independent of Ca^2+^ [41]. In addition, it interacts with CIRCADIAN CLOCK ASSOCIATED 1 (CCA1) and LATE ELONGATED HYPOCOTYL (LHY), two-morning circadian oscillator proteins [42,43]. The *CKB3*-overexpressing transgenic plants display shorter periods of *CCA1* and *LHY* and exhibit early flowering compared to wild-type in both LD and SD conditions [43].

Altogether, CaM, CPK, and Ca^2+^ transporters appear to regulate flowering, including a Ca^2+^-regulated Na^+^/Ca^2+^ exchanger AtNCL [44], a vacuolar H^+^/Ca^2+^ transporter and calcium exchanger1 (CAX1) [45], and a chloroplast-localizing calcium ion transporter ALBINO3/PPF1 [46,47]. AtNCL is a tonoplast-localized Ca^2+^ transporter with EF-hand domains and regulates Na^+^ and Ca^2+^ homeostasis. The *atncl* mutants are Ca^2+^ tolerant but sensitive to Na^+^ stress and show early flowering in LD due to high expression of *CONSTANS* (*CO*) and *FT* [44]. Another tonoplast Ca^2+^ transporter, CAX1, regulates flowering time. The *cax1-1* mutant delays flowering and is tolerant to Mg^2+^ and Mn^2+^ stresses but can survive Ca^2+^ deficiency stress [45]. Interestingly, AtNCL and CAX1 are both implicated in auxin signaling. Endogenous auxin accumulation is higher in *atncl* than in wild-type, while the auxin level is reduced in *cax1-1* [44,45]. *AtNCL* transcript is downregulated following the exogenous auxin treatment, whereas the root growth of *cax1-1* is insensitive to auxin treatment. However, how the intercellular Ca^2+^ homeostasis, controlled by vacuolar-localizing Ca^2+^ transporters and simultaneous auxin signaling, regulates flowering time is unclear. Another calcium transporter, PPF1/ALBINO3, which regulates flowering time, is localized at chloroplast sub-organelles [47]. Overexpression of *PPF1* causes higher Ca^2+^ accumulation in the chloroplasts, while *PPF1* antisense transgenic plants contain a low level of calcium ion. PPF1/ALBINO3 delays flowering by upregulating the expression of floral repressor *TERMINAL FLOWER1* (*TFL1*), without altering the *FLC* expression [46]. Antisense transgenic plants of *ALBINO3*, *ALBINO3(-)* display early flowering due to high expression of *LFY*, whose transcription is negatively regulated by TFL1 [46,48]. Experimental evidence suggests that calcium transporters and calcium-dependent kinases are involved in the flowering transition; however, the detailed molecular mechanism remains to be clarified. Moreover, it would be interesting to determine how the changes in subcellular Ca^2+^ storage capacity and cytosolic Ca^2+^ oscillation is related to flowering. Also, further investigation is required to identify the molecules that sense the changes and integrate and transfer the signals to flowering pathways.

### 2.2. Magnesium (Mg)

Magnesium participates in the growth and reproduction of plants [49], as Mg is critical for the conformational stabilization of macromolecules such as nucleic acids, proteins, and chlorophylls. It controls the activities of several enzymes such as kinases, H^+^-ATPases, and DNA and RNA polymerases. Thus, Mg deficiency in plants results in a reduced photosynthetic rate, chlorophyll degradation, ROS generation and oxidative damage [50], and the disruption of sucrose loading to the phloem [51,52]. Regarding flowering, several Mg^2+^ transporters (MGT) in Arabidopsis are known to be essential for pollen development. Disruption of specific Mg^2+^ transporters, mitochondria-localizing AtMGT5, plasma membrane-localizing AtMGT9, and ER-localizing AtMGT1 and AtMGT10 leads to pollen abortion and defective male gametophyte development and male fertility [53,54,55,56]. In addition, Mg^2+^-dependent exonucleases encoded by the nuclear genes, DEFECTIVE IN POLLEN ORGANELLE DNA DEGRADATION1 (DPD1; localized in plastids and mitochondria) and AtMGT20 (localized in mitochondria in developing pollen), degrade the paternal organelle DNA [50,57]. DPD1 homologs are present in angiosperm but not in moss, algae, or animals [58]. A previous study revealed that Mg^2+^ localizes in the shoot apex during flowering induction in *P. nil*, an SD flowering plant model [59], which is clear evidence of the involvement of Mg in flower development. During the vegetative stage or continuous light conditions, Mg^2+^ accumulates in the top layers of shoot apical meristem (SAM). However, an SD treatment reduces the Mg^2+^ content in the top layers and triggers floral induction. Mg is considered an important environmental factor in maintaining plant circadian rhythms because Mg deficiency causes longer periods of circadian oscillators, *CCA1* [60,61]. Thus, an altered circadian rhythm induced by Mg malnutrition in the soil might affect the photoperiodic flowering pathway in plants.

### 2.3. Sulfur (S)

Sulfur is a critical macronutrient for plants. As it is present in amino acids such as cysteine and methionine, it plays important roles in the activities of proteins and co-enzymes, prosthetic groups, and vitamins and antioxidants [62,63]. FIERY1 (FRY1), an inositol polyphosphate 1-phosphatase, involves sulfur metabolism by dephosphorylating 3′-phosphoadenosine 5′-phosphosulfate (PAPS) and converting it into adenosyl phosphosulfate (APS) [64,65]. Interestingly, it has a 3′(2′), 5′-bisphosphate nucleotidase activity, dephosphorylating 3’-phosphoadenosine 5’-phosphate (PAP) to produce adenosine monophosphate (AMP) and inorganic phosphate [64,65]. The loss-of-function mutants of *FRY1* possess pleiotropic phenotypes. FRY1 is involved in various biological processes such as sulfate assimilation [66], abscisic acid and stress signaling [67], cold and drought tolerance and signaling [68,69], RNA silencing and miRNA accumulation [70,71], fatty acid oxygenation regulation [72], oxidative stress response [73,74] leaf morphology [75], lateral root formation [76], circadian rhythm [77], and light-regulated responses, including hypocotyl and petiole growth, and flowering [78]. The *fry1* mutant exhibits late flowering due to the reduced *FT* transcript [78]. The major stress-related responses and physiological phenotypes of the *fry1* mutant could be due to the high accumulation of PAP, which goes to the nucleus from chloroplast as a retrograde signal during abiotic stress [68]. However, the molecular explanation for delayed flowering in *fry1* mutants remains to be investigated.

Glutathione (GSH), a cysteine-containing tripeptide (γ-glutamyl-cysteinyl-glycine), participates in homeostasis and cellular defense, including redox status, redox signal transduction, and detoxification of ROS and toxic compounds [79,80]. GSH biosynthesis from three constituent amino acids, L-glutamate, L-cysteine, and glycine, needs two ATP-dependent enzymes, γ-glutamyl-cysteine synthetase (γ-ECS or GSH1) and GSH synthetase (GS or GSH2). The plastid-localizing GSH1 catalyzes the first rate-limiting step to produce γ-glutamyl-cysteine (γ-EC). Plastid and cytosolic GSH2 sequentially adds glycine to γ-EC, generating GSH [81]. GSH increases plant tolerance to abiotic and biotic stresses [82,83,84] and delays senescence and flowering [85]. *GSH1* overexpressing transgenic plants show delayed flowering due to a high *FLC* transcript level and a low *FT* level. In contrast, *pad2-1* (*phytoalexin deficient 2-1*), a mutant allele of *GSH1*, shows an earlier flowering phenotype [85]. However, further studies are required to understand the molecular connection between the antioxidant enzyme and flowering time.

## 3. Micronutrients and Flowering Time

### 3.1. Boron (B)

Boron (B) is an essential micronutrient for plant growth and development as it is involved in cell wall assembly, maintenance of plasma membrane functions, developmental processes in root and shoot meristems, stimulation of reproductive tissues, improvement of seed quality, and biosynthesis of a few metabolic compounds such as antioxidants and polyphenols [86,87,88]. It also participates in carbohydrate metabolism and translocation, pollen germination and pollen tube developmental process, cell division, and indole-3-acetic acid (IAA) oxidase activity (increasing IAA amount) [86,87,88]. Thus, boron deficiency causes inhibition of root elongation and pollen tube growth, retarded growth with reduced leaf expansion, and fertility loss [89]. In contrast, boron is toxic to plants at high concentrations by interrupting cell division and limiting root growth [90] because the optimum boron concentration required for plant growth and development is narrow [91,92]. Boric acid and borate anions form complexes with cis-hydroxyl groups (-OH). One of the main functions of boron is to cross-link the pectic polysaccharide rhamnogalacturonan-II (RG-II) in the cell wall [93]. In the context of maintaining the cell wall structure and cell division, initial studies focused on the role of boron in pollen development and germination [87,88]. Boric acid channels exhibit distinct cell expressions such as plasma membrane localizing nodulin 26-like intrinsic proteins (NIPs), and borate transporters (BORs). Among them, AtNIP7;1 [94], AtNIP4;1, and AtNIP4;2 [95] are expressed in anther tapetum cells, pollen, and the pollen tube, respectively, participating in pollen germination and elongation. BOR1, predominantly expressed in anthers, is involved in pollen germination and elongation in rice [96]. 

Boron homeostasis appears to be involved in flower development because the sensitivity to boron deficiency is higher at the reproductive stage than the vegetative stage. A few boron transporters are expressed in shoot meristems and regulate floral development [89,97]. BRAHMA/BRM protein, degraded via 26S proteasome-mediated degradation upon high-boron stress, is involved in flowering and boron-tolerance by alleviating boron-induced DNA damage [98,99,100]. AtBRM, a chromatin-remodeling ATPase, is a constitutive subunit of an ATP-dependent SWItch/Sucrose Non-Fermentable (SWI/SNF) chromatin remodeling complex. It binds to acetylated lysine residue of histone tails through its bromodomains [100]. *AtBRM*-silenced and null mutants exhibit various defect phenotypes, including dwarf and small plants [99,100], altered leaf and root development with curled leaves [99,101], defects in floral organ patterning [99,102], hypersensitivity to ABA and drought tolerance [103], hypersensitivity to P deficiency [104], and boron tolerance [98]. In addition, the loss-of-function mutants of *BRM* show early flowering both in LD and SD due to high expression of floral activators, *CO, FT,* and *SUPPRESSOR OF OVEREXPRESSION OF CO 1* (*SOC1*) [99,100], and reduced expression of a floral repressor, *SHORT VEGETATIVE PHASE* (*SVP*) [105]. Interestingly, the late flowering of the *brm* mutant was due to the high expression of *FLC*, which is also a floral repressor [106]. BRM controls the *FLC* expression through the methylation status of histones on the *FLC* locus with reduced H3K4me3 and increased H3K27me3 levels, though it is not dependent on the vernalization pathway [106]. A genome-wide analysis using chromatin immunoprecipitation followed by next-generation sequencing (ChIP-seq) showed that the H3K27me3 level is significantly increased at the *SVP* locus in *brm* mutants compared to wild-type [105]. In addition, the *SVP* transcript is decreased in the *brm* mutant, suggesting that BRM promotes the expression of *SVP*, a floral repressor [105]. Moreover, the chromatin remodeler BRM physically interacts with a histone demethylase, RELATIVE OF EARLY FLOWERING 6 (REF6, an H3K27me3 demethylase) to facilitate transcriptional activation of target genes through the removal of the repressive histone mark H3K27me3. One of its target genes is *TARGET OF FLC AND SVP1* (*TFS1*), a transcription factor expressing on the flanks of SAM and promoting floral transition or induction both in LD and SD. BRM and REF6 complex is recruited to the *TFS1* loci in a SOC1-dependent manner to increase chromatin accessibility and promote *TFS1* transcription [107]. Recently, it has been reported that BRM protein interacts with a transcription factor GNC (GATA, NITRATE-INDUCIBLE, CARBON METABOLISM INVOLVED), and this complex directly binds to the *SOC1* genomic locus to suppress *SOC1* expression [108]. In addition, BRM appears to be involved in gibberellic acid (GA) signaling-mediated flowering by forming a complex with DELLA proteins (GA-negative regulators) and another transcription factor, NUCLEAR FACTOR Y-C (NF-YC). In normal conditions (without GA), DELLA protein promotes the interaction of BRM with NF-YC; thus, NF-YC is not able to bind to SOC1, resulting in late flowering. Meanwhile, DELLA-RBM binds to the *SOC1* gene locus, repressing the expression of *SOC1* through decreasing the H3K4me3 level at the *SOC1* chromatin. These processes subsequently prevent floral transition. However, GA induces the 26S proteasome ubiquitination-mediated degradation of DELLA proteins. Consequently, NF-YCs, transcriptional activators of *FT* and *SOC1,* become free from BRM protein, promoting flowering [109]. In addition, high boron increases histone acetylation in chromatin, inducing relaxed chromatin and consequent high susceptibility to double-strand DNA break. Meanwhile high boron degrades BRM protein, a chromatin-remodeling ATPase, reducing chromatin opening, which provide less chance for DNA damage [98]. These studies indicate that boron homeostasis and flowering (upon boron stress) implicate genetic regulations of autonomous and epigenetic regulation and hormonal response.

### 3.2. Zinc (Zn)

Zinc is an essential micronutrient in enzyme-mediated catabolic reactions, plant growth, and development. Zinc deficiency causes a significant reduction in carbohydrate and protein levels, which in turns restricts chlorophyll biosynthesis and leads to chlorosis in newly formed leaves. Suppression of stem elongation due to zinc deficiency reduces flowering and yield because of poor bud development [110]. High Zn concentration (not very high at the stress level) induces early flowering in *Arabidopsis arenosa* [111]. Supply of exogenous zinc to *Rhynchostylis retusa* L. promotes flowering induction, consistent with the previous study that high zinc concentration (750 ppm) induced early flowering [112]. A horticulture crop, gladiolus (*Gladiolus grandiflorus* L.), also exhibited early flowering after exogenous Zn and Fe application [113]. In addition to flowering time, applying Zn and Fe increases flower longevity, showing the synergistic effect of simultaneous application of Zn and Fe [113]. Applying micronutrient fertilizer containing Zn, B, Fe, and Mn shortens the days to first flower emergence in gerbera (*Gerbera jamesonii* L.) [114]. Application of zinc oxide (ZnO) nanoparticles (ZnO NPs) caused the promotion of plant growth, early germination and flowering, and a significant increase in fruit yield and seed weight in wheat, barley, and onion, indicating the positive impacts of Zn on overall plant growth and yield [115,116,117].

The zinc ion binds to amino acid residues and forms zinc finger (ZnF) motifs, which allow proper functionality and structural stability. Zinc finger proteins (ZFPs) with zinc finger motifs are classified into C_2_H_2_, C_2_HC, C_2_HC_5_, CCCH, C_3_HC_4_, and others according to the order and number of Zn-binding amino acid residues [118]. ZFPs play important roles in growth and development, abiotic stress response, and immunity against pathogens [119,120,121,122]. Various ZnF transcription factors belonging to distinct clades are involved in flowering regulation by modulating the transcription of floral regulators such as *FT*, *CO*, and *FLC*. Arabidopsis FILAMENTOUS FLOWER (FIL)/YABBY1 (YAB1) containing N-terminal CH_2_C_6_ ZnF domain and C-terminal YABBY domain participates in the formation of inflorescence and floral meristems by coordinating with AP1 and LFY [123,124]. Arabidopsis RING-domain or C_3_H_2_C_3_-type zinc finger protein, RED and FAR-RED INSENSITIVE 2 (RFI2), was reported to suppress the expression of *CO* and *FT* in the phytochrome B (phyB) signaling-mediated photoperiodic flowering pathway. The *rfi2-1* mutant showed increased *CO* and *FT* expression and early flowering under both LD and SD [125]. PhyB-interacting Zn transcription factors, VASCULAR PLANT ONE-ZINC FINGERs (AtVOZ1 and AtVOZ2), are positive regulators of a vernalization-induced flowering pathway, as the late-flowering phenotype due to elevated *FLC* transcript level in *voz1_voz2* double mutant is suppressed by vernalization [126,127,128]. 

Overexpression of Arabidopsis CCCH-type ZnF protein AtZFP1 leads to late flowering with decreased *FT* and *SOC1* and increased *FLC* and *CO* expression. Contrastingly, mutant plants show early flowering with increased *FT* and *SOC1* and decreased *FLC* and *CO*, indicating that AtZFP1 suppresses flowering by promoting *FLC* expression [129]. Additionally, AtZFP1 is involved in resistance to salt stress by elevating unsaturated fatty acid content [129]. The overexpression of alfalfa (*Medicago sativa*) CCCH-type ZnF, *MsZFN* in Arabidopsis exhibits delayed flowering with increased *FLC* levels and decreased *FT*, *SOC1*, and *GI* expression in LD [130]. However, Arabidopsis overexpressing another CCCH-type ZnF *ZFP3* from *Adonis amurensis* can flower at very cold temperatures. It also shows early flowering under normal and low temperatures due to higher expression of *FT* and lower *FLC* [131]. A double mutant *khz1khz2* in CCCH-ZnF and K-homolog domain KHZ1 and KHZ2 proteins exhibit late flowering with high expression of *FLC* [132]. In rice (*Oryza sativa*), an SD flowering plant, *Early heading date 4* (*Ehd4*) encoding CCCH-type ZnF protein promotes the expression of *Heading date 3a* (*Hd3a),* a rice counterpart of Arabidopsis *FT* and *RICE FLOWERING LOCUS T* (*RFT1,* a homolog of *Hd3a*) in the *Ehd1*-dependent manner, which is a rice-specific floral inducer in SD [133,134]. 

Arabidopsis C_2_H_2_-type ZnF factor, LATE FLOWERING (LATE), participates in the photoperiod pathway of FT. The overexpression of *LATE* delays flowering with impaired inflorescence growth and sterile and incomplete flower formation [135]. LATE can interfere with the activity of *FT* upstream regulatory factors to block photoperiodic flowering since *LATE* expression in leaf vasculature suppresses *FT* expression in LD. However, whether LATE regulates *FT* by directly binding to the *FT* promoter locus or through other *FT* repressor complexes is unknown. The plant-specific C_2_H_2_-type ZnF-SET domain protein, AtCZS, functions as a repressor of *FLC* through chromatin remodeling in Arabidopsis. The *CZS* mutation causes hyperacetylation of histone H4 and reduces demethylation of histone H3 in the *FLC* locus, consequently leading to increased *FLC* and delayed flowering [136]. However, another C_2_H_2_ ZnF protein with a proline-rich domain, SUPPRESSOR OF FRIGIDA 4 (SUF4), activates *FLC* transcription by binding to *the FLC* promoter region after forming a complex with FRIGIDA (FRI) and FRI-LIKE 1 (FRL1) [137,138]. *suf4* mutants strongly suppress the late flowering of *FRI* mutants with reduced H3K4 trimethylation at *FLC* [137,139]. SUF4 may recruit histone H3 methyltransferase EFS (EARLY FLOWERING IN SHORT DAYS), a flowering inhibitor, and the PAF1-like complex to the *FLC* locus [140]. Another rice FCS-like divergent C_2_H_2_ ZnF group protein, FCS-LIKE ZINC FINGER 2 (OsFLZ2), destabilizes an SD floral activator, OsMADS51, to repress its transcriptional activity, resulting in lower expression of *Early heading date 1* (*Ehd1)* [141]. 

Maize ZnF INDETERMINATE 1 (ZmID1) was suggested to act as a transcriptional regulator of the floral transition in a non-cell-autonomous manner due to late flowering in the *id1* mutant [142,143]. Rice EARLY HEADING DATE 2 (Ehd2), an ortholog of maize ZmID1, was later called Rice Indeterminate 1 (RID1), a C_2_H_2_-type ZnF transcription factor. It promotes flowering by positively regulating *Ehd1* in SD conditions, showing the mutation in Ehd2 causes extremely late flowering with decreased transcription levels of *Ehd1*, *Hd3a* (an ortholog of *FT*), and *RFT1* [144]. SUPPRESSOR OF RID1 (SID1)/*Oryza sativa* INDETERMINATE DOMAIN 4 (OsIDD4) is identified as a suppressor of the never-flowering phenotype of *rid1* [145]. SID1, an IDD family member, induces *Hd3a* and *RFT1* expression by directly binding to the *Hd3a* and *RFT1* promoter region [145]. The conserved IDDs of OsIDD4, OsIDD1, and OsIDD6 also rescue the never-flowering phenotype of *rid1*, indicating that OsIDD1 and OsIDD6 redundantly function in the regulation of flowering time with SID1/OsIDD4 [145]. 

A few ZnF proteins with histone demethylase activity can participate in histone modification of *FLC* due to the presence of the JmjC-domain. EARLY FLOWERING 6 (ELF6)/JmjC DOMAIN-CONTAINING PROTEIN 11 (JMJ11) and RELATIVE OF EARLY FLOWERING 6 (REF6)/JMJ12 are plant-specific C_2_H_2_-type ZnF-containing Jumonji C-terminal (JmjC) proteins. These proteins modulate the demethylase activity of H3K27me3. The *ref6* mutant is late flowering because of high *FLC* expression, while the *elf6* mutant is early flowering due to the downregulated *FLC* and sequential upregulation of *FT* [146]. ELF6/JMJ11 binds to the *FLC* locus to activate its expression [147]. Also, REF6/JMJ12 associates with *SOC1* locus to promote *SOC1* expression in LD [148]. Another H3K27me3 demethylase, JMJ13, contains a C_4_HCHC-type ZnF domain and is a repressor of flowering time in response to photoperiod and high temperature. The *jmj13* mutant is early flowering under the LD at normal (22 °C) and high (28 °C) temperatures, and under the SD at high temperatures but not at normal temperatures due to reduced expression of floral repressor gene *SHORT VEGETATIVE PHASE* (*SVP*) [149]. Rice Se14 containing JmjC domain and C_2_H_2_-type zinc finger motifs exhibits upregulated *RFT1*, a florigen-like gene, and increases trimethylated H3K4 in the *RFT1* promoter region, resulting in early flowering [150]. JMJ14, another H3K4 demethylase with a C_5_HC_2_-type ZnF domain from Arabidopsis, represses floral transition but does not regulate *FLC* expression. Nevertheless, it inhibits the expression of several floral integrators such as *FT*, *SOC1*, *AP1*, and *LFY* [151,152,153]. Additionally, JMJ14 forms a complex with EMBRYONIC FLOWER 1 (EMF1, a component of polycomb group (PcG) complex) and LIKE HETEROCHROMATIN PROTEIN1 (LHP1). This complex binds to the *FT* locus and represses the expression of *FT* from dawn until dusk and at night, whereas the occupation of this complex on *FT* promoter is alleviated to promote *FT* transcription at dusk [154]. However, two H3K4 demethylases with a C_5_HC_2_-type ZnF domain, JMJ15 and JMJ18, reduce H3K4me3 at the *FLC* locus, thus inducing flowering with repressed *FLC* and increased *FT* transcript levels [155,156,157]. 

CO/B-Box domain protein 1 (BBX1), a well-characterized ZnF protein in flowering regulation, possesses a B-box characterized with a zinc finger binding domain with conserved cysteine and histidine residues. BBX1 is a positive flowering regulator promoting *FT* expression [158]. Including the *CO* gene, Arabidopsis *CO-LIKE 9* (*COL9*)/*BBX7*, *COL5*/*BBX6*, and *EMF1-INTERACTING PROTEIN 6* (*EIP6*)/*BBX32* are involved in the regulation of flowering time [159,160,161]. The COL3/BBX4 and BBX32 complex activates *FT* transcription by binding to the *FT* promoter region, initiating floral transition [162]. A rice ortholog of Arabidopsis CO, HEADING DATE 1 (Hd1) with two B-boxes is also well known as a central integrator of the SD-photoperiod-induced flowering pathway in rice. On the other hand, *OsCOL3* with a single B-box suppresses flowering, exhibiting decreased *FT-LIKE* (*FTL*) expression and late flowering in *OsCOL3*-overexpressing plants under SD conditions [163,164]. The suppression of *CmBBX24,* a *Chrysanthemum morifolium* ZnF transcription factor, leads to early flowering and increased photoperiod- and GA biosynthesis-associated gene expression. In contrast, the *CmBBX24* overexpression causes late flowering and decreased photoperiod- and GA biosynthesis-associated gene expression, suggesting that *CmBBX24* might regulate flowering via both photoperiod and gibberellin biosynthesis pathways [165].

ZFPs are present in approximately 0.8% of the Arabidopsis genome [166]. They contribute to overall plant development, including flowering induction and floral morphogenesis through transcriptional and chromatin regulation. However, the exact role of zinc in ZFP activation and action has not been elucidated completely. Thus, further work on flowering induction and ZFP modification according to Zn status will help interpret the critical roles of zinc in the functional activation of ZFPs and the regulation of plant development.

### 3.3. Iron (Fe) 

As mentioned in Section 3.2, several JmjC domain proteins with demethylase activities modulate flowering time by regulating *FLC* expression through histone remodeling. These JmjC domain histone demethylases require Fe^2+^ and α-ketoglutarate as cofactors [167]. JMJ30 directly binds to the *FLC* locus, enhances *FLC* transcription by demethylating histone H3K27me or reducing the level of H3K27me3, an epigenetic *FLC*-silencing marker, and suppresses flowering. The double mutant of JMJ30 and its homologous JMJ32, *jmj30_jmj32,* shows early flowering under vernalized conditions with a high accumulation level of H3K27me3 and a reduced level of *FLC* transcription [168,169]. JMJ27 also shows delayed flowering by demethylating H3K9 in the promoter regions of major flowering regulators such as *FLC*, *CO*, *FT*, and *SOC1* [155,170,171,172].

In addition to being a cofactor of histone methylase enzymes, iron (Fe) is an essential micronutrient because it is required for redox reaction, electron transport chain, chlorophyll biosynthesis and photosynthesis, and nitrate and sulfate reduction [173]. Fe, one of the most abundant elements in soil, is absorbed in the form of ferrous (Fe^2+^, reduced state) by plants. Fe uptake, translocation, distribution, compartmentalization, and storage are important processes for Fe homeostasis [174]. Several genes in Fe homeostasis (Fe transport and storage) are circadian regulated [175], and Fe deficiency lengthens the periods of circadian oscillations. Fe-dependent circadian rhythms are lost in *GIGANTEA* (*GI*) mutants, a circadian clock gene regulating photoperiodic flowering, or ZEITLUPE (ZTL). ZTL is a component of the circadian clock and an interactor of GI protein [175,176]. Moreover, the longer period upon Fe deficiency is impaired in Fe uptake mutants of *IRON-REGULATED TRANSPORTER 1* (*IRT1*) and *FER-LIKE IRON DEFICIENCY INDUCED TRANSCRIPTION FACTOR* (*FIT*) [175,176,177], suggesting that Fe nutrient homeostasis works as a feedback signal for plant circadian rhythms and vice versa.

Plants have developed regulatory systems to maintain Fe uptake and homeostasis. Strategy I in dicot and non-graminaceous monocots adapts a reduction approach, secreting acidic solution via H^+^-ATPase2 (AHA2) and reducing Ferric (Fe^3+^) to Fe^2+^ by FERRIC REDUCTION OXIDASE 2 (FRO2), which is a Fe^3+^ chelate reductase. The reduced and soluble Fe^2+^ is transported via IRT1 at the plasma membrane of root epidermal cells [174]. Strategy II in grass monocot utilizes siderophores secreted to the rhizosphere to chelate Fe^3+^. The chelated Fe complex is transported via YELLOW STRIPE1 (YS1) transporters [174]. Iron deficiency induces significant changes in the expression of genes involved in Fe uptake. The basic helix–loop–helix (bHLH) proteins work as key regulators in the Fe deficiency-induced transcriptional regulatory network [178,179]. In Arabidopsis, which exploits strategy I for Fe uptake, at least 17 bHLH transcription factors from 6 subfamilies are involved in Fe homeostasis [174]. *FIT/bHLH29*, specifically localized in roots, is induced under Fe deficiency. The expression of many Fe-regulated genes, including *FRO2*, is dependent on FIT/bHLH29.

Arabidopsis bHLH38, bHLH100, and bHLH101 regulate Fe homeostasis by transcriptional activation of Fe uptake-associated genes, *IRT1* and *FRO2* [180]. The *bhlh38 bhlh100 bhlh101* triple mutant shows severe chlorosis, reduced Fe contents, early flowering, and increased *FT* expression [181,182]. Previous reports suggest that Fe deficiency suppresses flowering via bHLH38-bHLH100-bHLH101-dependent mechanisms [181,182]. In addition, FIT/bHLH29 and its interactors bHLH38 and bHLH39 physically interact with GA-negative regulator DELLA proteins [183]. FIT/bHLH29 and DELLA heterodimer complex do not inhibit the heterodimer formation of FIT and Ib bHLH proteins (a subgroup including bLHL38, 39, 100, and 101); however, *FIT* transcription is inhibited by preventing the binding of the heterodimer (FIT and Ib bHLH) to the *FIT* promoter locus [183]. Furthermore, in response to Fe deficiency, DELLA proteins accumulate in the root meristem to inhibit root growth but are degraded in the epidermal cells of the root differentiation zone. Further, the FIT protein released from the DELLA-FIT complex activates Fe uptake, while DELLA proteins are distributed in all root tissues in normal conditions, suggesting that GA signaling is involved in Fe deficiency response through the interaction of DELLA and FIT [183]. GA signaling through DELLA proteins regulates flowering time, as DELLAs repress the expression of *LFY* and *SOC1* (floral meristem identity genes), especially in SD [184]. As a superfamily containing at least 162 members in Arabidopsis, bHLH proteins participate in mineral nutrient and abiotic stress responses as well as in developmental processes such as flowering and hormone signaling [179]. Further studies on Fe’s roles in the functional activation of bHLH proteins will enrich our understanding of Fe homeostasis and plant flowering, which would help improve plant productivity. 

Iron is an important enzyme catalyst, especially in electron transport chains. Iron-sulfur (Fe-S) is a type of catalytic iron center (or cofactor) hemes. A central member of the cytosolic Fe-S cluster, MMS19/MET18, is reported to increase the efficiency of Fe-S cluster assembly on DNA polymerase. The Fe-S cluster of MET18 is required for DNA glycosylase ROS1-mediated DNA methylation [185,186]. Interestingly, the *mms19* mutant shows early flowering due to the upregulation of *FT*, *AP1,* and *AP2* and the downregulation of *FLC* [185]. The *met18* mutant shows DNA hypermethylation at hundreds of genetic loci and subsequently induces transcriptional silencing of those genes. However, the methylation in transposable element (TE) regions is reduced in the *met18* mutant [187]. Thus, DNA methylation status changed by MMS19/MET18 and its interacting proteins seems to influence the expression of genes associated with flowering, though clear molecular analysis remains to be elucidated.

### 3.4. Copper (Cu)

Copper is an essential redox-active transition metal, which changes its oxidation status between cuprous (Cu(I), Cu^+^, reduced form) and cupric (Cu(II), Cu^2+^, oxidized form) and acts as a structural element in regulatory proteins [188,189]. While Cu deficiency can negatively affect developmental processes such as photosynthesis, respiration, and reproductive success, copper excess causes phytotoxicity because free Cu ions can produce toxic hydroxyl radicals, which can damage macromolecules and bind to a thiol (or sulfhydryl) group [190,191]. Cu homeostasis is tightly regulated by Cu transporters and metallo-chaperones [188,189].

Copper participates in photosynthetic electron transport, mitochondrial respiration, oxidative stress responses, lignification of cell walls, and ethylene hormone signaling [188], as it serves as a cofactor of plastocyanin, an electron carrier (between the cytochrome-b_6_f complex and photosystem I located in the thylakoid membrane) in the thylakoid lumen [192,193,194], cytochrome c oxidase in the mitochondrial inner membrane [195], copper/zinc superoxide dismutase (Cu/ZnSOD) in stroma and cytosol [196], laccase and amine oxidase in the apoplast [197,198], polyphenol oxidase in the thylakoid lumen [199,200], and phytocyanin (or plantacyanin) in the apoplast [201,202]. In addition, Cu is involved in developing male and female reproductive organs, affecting grain, seed, and fruit formation [203]. Phytocyanins, another group of Cu proteins, are secreted to extracellular matrices of pollen and pistil and are involved in pollen tube guidance [201,204]. In addition, the distribution or reallocation of Cu seems to regulate the development of reproductive organs. Among six Cu transporters (CTR) in Arabidopsis, Copper Transporter Protein 1 (COPT1) [205] and COPT6 [206,207] are mostly expressed in the root tips and pollen and at the vasculature of green tissues and reproductive organs, respectively, to facilitate Cu redistribution under its scarcity.

SQUAMOSA PROMOTER BINDING PROTEIN-LIKE7 (SPL7) is a key regulator of Cu homeostasis [208,209]. Cu^+^ binds to the highly conserved cysteine and histidine residues in SPL7 [209,210], though Cu does not change the expression of *SPL7* at the transcriptional level. During Cu deficiency, it binds to GTAC promoter sequences in copper response elements (CuREs) of target genes and activates the transcription of several genes involved in copper homeostasis [208,209], indicating the possibility of post-translational modification in the SPL7 protein in response to Cu availability. SPL7 directly regulates Cu transporters and chaperons such as COPT1, COPT2, COPT6, and Yellow Stripe-Like2 (YSL2, a transporter of nicotianamine-metal complexes) [202,209]. Additionally, it activates specific microRNA transcription, including miRNA397, miRNA398, miRNA409, and miRNA857 [209,211,212,213]. The Cu-miRNAs inhibit the expression of Cu proteins such as phytocyanin, laccase, and Cu/ZnSOD, thereby regulating Cu allocation and homeostasis [214,215]. The *spl7-1* mutant exhibits lightly delayed flowering both in Cu-deficient medium and normal conditions, probably due to high miRNA156 accumulation, whose overexpression in Arabidopsis and other plants delays flowering and prolongs the juvenile phase [216]. miRNA156 expresses highly in the juvenile phase, and its expression reduces before the transition to flowering and controls the vegetative-to-reproductive transition [217,218]. Limited Cu conditions delay flowering time with reduced expression of the floral activator *FT* [219]. However, further study is required to understand how Cu deficiency also induces miRNA172 accumulation, which promotes flowering through the posttranscriptional regulation of floral organ identity gene *APETALA2* (*AP2*), AP2-like transcriptional factors, including *TARGET OF EAT1* (*TOE1*), *TOE2*, *TOE3*, *SCHLAFMUTZE* (*SMZ*), and *SCHNARCHZAPFEN* (*SNZ*), and floral repressors and functions upstream of *FT* in a CO-independent manner [219,220,221,222,223].

The excess of copper during heavy metal treatment (concentrations above 50 μM and 100 μM), such as copper sulfate (CuSO_4_), induces ethylene production [224]. Ethylene causes delayed flowering via an enhanced accumulation of DELLA proteins, which represses floral meristem identity genes *LFY* and *SOC1* [225] and induces ethylene signaling-associated chromatin remodeling to promote the expression of *FLC*, a floral repressor [226]. Moreover, ethylene activates downstream signaling through ETHYLENE RESPONSE FACTOR 1 (ERF1), which binds to the *FT* promoter and delays flowering time by reducing *FT* expression [227]. It appears that the ethylene-response pathway is regulated by copper homeostasis. Arabidopsis heavy metal transporting ATPase 7 (HMA7)/RESPONSIVE TO ANTAGONIST 1 (RAN1), a P-type ATPase Cu transporter that is localized in the endomembrane reticulum (ER) of root and flower tissues, is required to deliver Cu to the ER-localizing ethylene receptors including Ethylene Response 1 (ETR1). The Cu binding to ETR1 is also needed for ethylene binding and functional receptors [228,229,230,231,232,233]. Two weak alleles of *ran1* mutants (*ran1-1* and *ran1-2*) only show ethylene phenotypes upon the treatment of trans-cyclooctene, an ethylene-response antagonist, but not to ethylene, and the phenotype is suppressed by adding Cu ions [228]. Two strong mutant alleles of *RNA1* (*ran1-3* and *ran1-4*) [229,234] exhibit ethylene responses, including short root and hypocotyl, radial expansion of the hypocotyl, and exaggerated curvature of the apical hook. These phenotypes correspond to the loss-of-function phenotypes of ethylene receptor mutants. Thus, Cu-limited conditions and excess amount up to the toxic level can both alter flowering time or affect reproduction, suggesting that Cu homeostasis influences flowering.

### 3.5. Manganese (Mn)

Manganese is an essential element as an enzyme cofactor in several biological processes, including photosynthesis, lipid biosynthesis, and oxidative stress response, although excessive Mn is toxic to plants [235,236,237]. Mn is interchangeable with other divalent cations such as Ca, Cu, Mg, and Zn, but Mg mostly replaces Mn because soluble Mn^2+^ (reduced form) is rapidly oxidized to plant-unavailable Mn oxides, particularly in high pH soil. In addition, Mg^2+^ is present 50 to 100 times more in the cell [235,236]. Mn is an important cofactor for enzymes involved in various biochemical reactions such as RNA polymerases, terpene synthases in isoprenoid biosynthesis, phenylalanine ammonia-lyase (PAL) in lipid biosynthesis, decarboxylases in the Kelvin Cycle, and various Golgi-localized glycosyl transferases [235,237,238]. However, Mn is indispensably required as a catalyst in the oxygen-evolving complex (OEC) of photosystem II (PSII), which splits water (H_2_O) into oxygen (O_2_) and produces electrons (light-induced water oxidation or photolysis of water, H_2_O → 2H^+^ + 2e^−^ + 1/2 O_2_) [239]. In addition, it serves as a cofactor of an antioxidant Mn superoxide dismutase (MnSOD, MSD) in mitochondria [196,240] and oxalate oxidase, which is secreted to apoplast catalyzing oxygen-dependent degradation of oxalate to CO_2_ and H_2_O_2_ and is involved in the pathogen defense response [241,242]. Various Mn transporters and ion chaperones are involved in Mn tissue and subcellular allocation and Mn homeostasis [235,236]. Thus, Mn deficiency or incorrect position of Mn results in a low photosynthetic rate and higher susceptibility to biotic and abiotic stresses [235]. Regarding reproductive development, Mn deficiency is reported to cause poor tasseling and to reduce anther development and pollen grain germination in maize [243]. Also, Mn homeostasis in the Golgi apparatus and protein glycosylation in Golgi seem important for pollen tube growth because the loss-of-function mutant of a cis-Golgi-localized Mn transporter, *PML3* (photosynthesis-affected mutant 72-like3), exhibits impaired pollen tube growth and Golgi glycosylation [244,245]. Mn cation is required for Golgi-localized transferases for protein *N*-glycosylation and cytoplasmic UDP-glycosyl transferases (UGT) transferring a glycosyl moiety from UDP-sugars (uridine 5′-diphospho sugar) to various small acceptor molecules during secondary metabolites and phytohormone biosynthesis [246,247,248]. UGT87A2, an Arabidopsis UGT, regulates flowering time by reducing the expression of *FLC*, a floral repressor. The *ugt87a2* mutant exhibits late flowering both in LD and SD with a high transcription level of *FLC* [249]. In addition, Arabidopsis MTM1 and MTM2, homologs of yeast MTM1 (manganese trafficking factor for mitochondrial SOD2), which activates mitochondrial MnSOD1 (MSD1) [250,251], regulate flowering time. The *mtm1 mtm2* double mutant shows early flowering than wild-type and both single mutants [252]. However, the underlying mechanism of how these Mn-dependent proteins regulate flowering time in plants needs to be depicted further.

### 3.6. Molybdenum (Mo)

Molybdenum (Mo) is an essential element for plants, though only a minute amount is required because it serves as the active site for Mo-requiring enzymes (molybdoenzymes) involved in different redox reactions in plants. It is biologically inactive until it forms a molybdenum cofactor (Moco), a complex with proteins [253,254]. More than 50 proteins are predicted to contain Mo; however, only 5 molybdoenzymes are characterized in plants, including sulfite oxidase (which catalyzes the conversion of sulfite to sulfate, the final step in the oxidative degradation of sulfur-containing amino acids cysteine and methionine) [255], xanthine dehydrogenase (which catalyzes the oxidation of purines) [256], aldehyde oxidase [which oxidizes a range of aldehydes to their corresponding carboxylic acid and is critical for the biosynthesis of phytohormones such as abscisic acid (ABA) and indoleacetic acid (IAA)] [257], nitrate reductase (NR) [which catalyzes the first reaction in nitrate assimilation, the reduction of nitrate (NO_3_^−^) to nitrite (NO_2_^−^)] [258], and (mitochondrial) amidoxime reducing component ((m)ARC) (which reduces N-hydroxylated compounds) [259]. Thus, the symptoms of Mo deficiency are similar to those of loss of function of Mo transporters and carrier proteins, impaired assembly of Moco, and defective molybdoenzymes [254,260,261,262].

Regarding flowering time, Mo-deficient medium induced flowering in a free-floating aquatic plant, duckweed (*Lemna paucicostata*), independently from daylength, and the nitrate reductase (NR) activity was inhibited [263]. The NR activity was reduced during flowering and seed development in soybean (*Glycine max*) [264]. Molybdoenzymes, nitrate reductase (NR), and amidoxime-reducing component (ARC) are involved in nitrogen assimilation producing nitric oxide (NO), an important signaling molecule [259,265]. Nitrate delays the flowering time at SAM through the master regulators of nitrate signaling, NIN-LIKE PROTEIN (NLP6) and NPL7, and by repressing *SOC1*, *SPL3* (*SQUAMOSA PROMOTER BINDING PROTEIN-LIKE3*), and *SPL4* [266,267]. A mutant overproducing NO, *nox1*, shows delayed flowering, while a mutant with less NO, *nos1*, exhibits early flowering [268]. It turns out that NO delays flowering by downregulating *CO* and *GI* and their downstream *LFY* and by increasing *FLC* [268,269]. Also, a low amount of NO in the NR double mutant *nia1 nia2* shows early flowering [270,271]. Notably, NO increases the amplitudes of circadian transcripts of *CRYPTOCHROME1* (*CRY1*), *LHY*, *CCA1,* and *TIMING OF CAB EXPRESSION 1* (*TOC1*), but decreases the amplitudes of the circadian transcripts of *CO* and *GI* [269]. In addition, CO and GI proteins (but not other clock oscillators) were increased following enhanced NO level [269], suggesting that NO is involved in light and circadian regulation of flowering. Thus, Mo appears to participate in flowering time via nitrogen assimilation. 

## 4. Perspectives and Conclusions

Recent studies have characterized the transcriptional regulation of floral-inducing or floral-identifying factors by transcription factors and enzymes. Secondary nutrients such as calcium, magnesium, and sulfur play vital roles in flowering and flower development through their involvement in various signal transduction pathways and the regulation of enzyme activities in diverse metabolisms. Additionally, these studies have underscored the importance of micronutrients as cofactors, as well as the roles of ion channels and transporters in plant growth and reproduction (Table 1). Although we suggested transcription factors and enzymes which possess microelement binding domains such as ZnFs and bHLHs, it is unclear how activities or stabilities of these proteins are altered under the microelement deficiency or excess. Additionally, due to the dosage-dependent photo-toxicity and the requirement of lower amounts for optimal plant nutrition, the biological and physiological mechanisms underlying the plant growth and flowering response to micronutrients are poorly understood. Specifically, there is a lack of knowledge regarding the conformational changes of proteins by microelements and the biochemical and molecular mechanisms that directly promote plant growth and flowering through micronutrients. 

This review aims to comprehensively summarize various transcription factors and enzymes that rely on micronutrients as cofactors for their functional activation or catalytic activity enhancement, including ZnFs, bHLHs, and JMJs (Table 1). Further research is required to uncover the specific roles of micronutrients in functional activation of microelement-containing proteins and to establish the correlation between micronutrient homeostasis and flowering regulation. The studies on essential micronutrients and their influence on regulating flowering time will enhance our understanding of nutrient-mediated floral induction and floral regulatory mechanisms. The outcomes of these studies would help researchers plan potential strategies for crop breeding and productivity improvement as a long-term goal because control of flowering time in agricultural and horticultural plants is crucial for pollinator visitation, disease and pest management, and ultimately, crop yield and quality.

## Figures and Tables

**Table 1 plants-12-03299-t001:** A summary of transcription factors and enzymes relying on micronutrients as cofactors for their functional activation or catalytic activity enhancement.

Plant	Gene Name	Protein Function	Element	Effect	Description	Reference
Arabidopsis	*CML24*	Calmodulin-like protein	Calcium	Promotion	*cml24* mutant delays flowering in LD with high expression of *CO*.	[34,35]
Arabidopsis	*CML23* and *CML24*	Calmodulin-like protein	Calcium	Promotion	*cml23 cml24* double mutant delays flowering in LD through increased *FLC*.	[34,35]
Arabidopsis	*CPK32*	Kinase	Calcium	Promotion	*cpk32* exhibits delayed and late flowering. CPK32 protein interacts with FCA and phosphorylates it in a Ca^2+^-dependent manner.	[38]
Arabidopsis	*CBK3/CRK1*	Kinase	Calcium	Promotion	*CKB3*-overexpression transgenic line exhibits early flowering in both LD and SD with shorter periods of circadian rhythm of CCA1 and LHY.	[42,43]
Arabidopsis	*NCL*	Na^+^/Ca^2+^ exchanger	Calcium	Inhibition	*NCL* overexpressing transgenic plants delays flowering with less expression of *CO* and *FT*, and *atncl-1* mutant flowers earlier in LD, not SD, due to high expression of *CO* and *FT*.	[44]
Arabidopsis	*CAX1*	Vacuolar H^+^/Ca^2+^ transporter	Calcium	Promotion	*cax1* displays late flowering.	[45]
Arabidopsis	*ALBINO3/PPF1*	Chloroplast-localizing calcium ion transporter	Calcium	Inhibition	PPF1 overexpressing plants flower late, while PPF1 antisense transgenic plants flower earlier.	[46,47]
Soybean	*GmCDPK38*	Kinase	Calcium	Promotion	*gmcdpk38* mutant flowers later and alters the expression of several flowering genes.	[39]
Tobacco	*NtCBK1*	Kinase	Calcium	Inhibition	Transgenic tobacco overexpressing *NtCBK1* exhibits late flowering.	[40]
Arabidopsis	*GSH1*	Gamma-glutamyl cysteine synthase	Sulfur	Inhibition	*GSH1* overexpressing transgenic plants exhibit delayed flowering due to high levels of *FLC* and low *FT* expression, while *pad2-1*, a *GSH1* mutant allele, flowers earlier than wild-type.	[85]
Arabidopsis	*FRY1*	3’(2’),5’-bisphosphate nucleotidase	Sulfur	Promotion	*fry1* mutant flowers late due to reduced *FT* expression.	[78]
Arabidopsis	*FIL*	ZnF TF	Zinc	Inhibition	*fil* mutant shows early flowering.	[123,124]
Arabidopsis	*LATE*	ZnF TF	Zinc	Inhibition	*LATE* ectopic expression results in late flowering.	[135]
Arabidopsis	*AtVOZ1* and *AtVOZ2*	ZnF TF	Zinc	Promotion	*voz1voz2* double mutant exhibits severely delayed flowering due to increased *FLC* transcription level.	[127]
Rice	*FLZ2*	ZnF TF	Zinc	Inhibition	*FLZ2* overexpressing plants display late flowering, and *flz2* mutant exhibits early flowering, showing alterations in floral integrator gene expression.	[141]
Alfalfa	*MsZFN*	Putative znf TF	Zinc	Inhibition	*MsZFN* overexpression in Arabidopsis causes late flowering under LD.	[130]
Maize	*ID1*	Transcription factor, IDD family	Zinc	Promotion	*id* mutant exhibits late flowering.	[142]
Arabidopsis	*CO*	ZnF TF, BBX family	Zinc	Promotion	*co* mutant exhibits late flowering.	[158]
Arabidopsis	*COL3/BBX4*	ZnF TF, BBX family	Zinc	Inhibition	*col3* exhibits early flowering with increased *FT* expression. COL3/BBX4 interacts with BBX32.	[162]
Arabidopsis	*COL5/BBX6*	ZnF TF, BBX family	Zinc	Promotion	Overexpression of *COL5/BBX6* promotes flowering.	[159]
Arabidopsis	*COL9/BBX7*	ZnF TF, BBX family	Zinc	Inhibition	*COL9* overexpressing plants display delayed flowering, while suppression of *COL9* flowers earlier.	[160]
Arabidopsis	*EIP6/BBX32*	ZnF TF, BBX family	Zinc	Inhibition	*eip6/bbx32* mutant displays early flowering, while overexpressing plants exhibit late flowering.	[161]
Rice	*HD1*	ZnF TF, BBX family	Zinc	Promotion	*hd1* exhibits late flowering.	[163]
Rice	*OsCOL3*	ZnF TF, BBX family	Zinc	Inhibition	*OsCOL3* overexpression causes late flowering through decrease in *FTL* expression.	[164]
Chrysanthemum	*BBX24*	ZnF TF, BBX family	Zinc	Inhibition	Suppression of *BBX24* causes early flowering via photoperiod and GA biosynthesis pathways.	[165]
Arabidopsis	*JMJ11/ELF6*	Histone methyltransferase, znf and jmjc	Zinc, iron	Inhibition	*elf6* mutant is early flowering due to the downregulation of *FLC* and sequential upregulation of *FT.*	[146,147]
Arabidopsis	*JMJ12/REF6*	Histone methyltransferase, znf and jmjc	Zinc, iron	Promotion	*ref6* mutant is late flowering because of high *FLC* expression. And REF6 /JMJ12 associates with SOC1 locus to promote *SOC1* expression in LD.	[146,148]
Arabidopsis	*JMJ13*	Histone methyltransferase, znf and jmjc	Zinc, iron	Inhibition	*jmj13* mutant is early flowering both in LD and SD.	[149]
Arabidopsis	*JMJ14*	Histone methyltransferase, znf and jmjc	Zinc, iron	Inhibition	*jmj14* mutant shows increase in *FT*, *SOC1*, *AP1*, and *LFY* expression and early flowering. JMJ14 interacts with EMF1, a component of a polycomb group complex, and suppresses *FT* transcription and flowering.	[151,152,154]
Arabidopsis	*JMJ15*	Histone methyltransferase, znf and jmjc	Zinc, iron	Promotion	*JMJ15* overexpression results in early flowering and repression of *FLC* level and reduced H3K4me3 at the *FLC* locus, resulting in increased *FT* expression.	[157]
Arabidopsis	*JMJ18*	Histone methyltransferase, znf and jmjc	Zinc, iron	Promotion	*JMJ18* overexpression results in early flowering and repression of *FLC* level and reduced H3K4me3 at the *FLC* locus, resulting in increased *FT* expression.	[156]
Arabidopsis	*JMJ27*	Histone methyltransferase, jmjc domain	Iron	Inhibition	*jmj27* mutant displays an increase in *CO*, *FT*, and *SOC1* expression and a decrease in *FLC* expression, resulting in early flowering.	[172]
Arabidopsis	*JMJ30* and *JMJ32*	Histone methyltransferase, jmjc domain	Iron	Inhibition	*jmj30 jmj32* shows early flowering under vernalized conditions with a high accumulation level of H3K27me3 and less *FLC* transcription.	[168,169]
Arabidopsis	*bHLH38, bHLH100,* and *bHLH101*	bHLH transcription factors	Iron	Inhibition	*bhlh38 bhlh100 bhlh101* triple mutant shows early flowering.	[181]
Arabidopsis	*MMS19*	A component of Fe-S cluster	Iron, sulfur	Inhibition	*mms19* mutant shows reduced Fe contents and early flowering.	[185]
Arabidopsis	*BRM*	Chromatin-remodeling atpase with bromodomain	Boron	Inhibition	The protein accumulation of BRAHMA is reduced through proteasomal degradation. *BRM*-silenced and loss-of-function mutation in *BRM* exhibits early flowering under both long days and short days.	[98,99,100]
Arabidopsis	*SPL7*	SQUAMOSA PROMOTER BINDING and transcription factor	Copper	Promotion	*spl7-1* mutant exhibits lightly delayed flowering both in Cu-deficient medium and normal conditions, probably due to high accumulation of *miRNA156.*	[216]
Arabidopsis	*MTM1* and *MTM2*	MnSOD1 activator	Manganese	Inhibition	*mtm1 mtm2* double mutant flowers earlier than wild-type and single mutants.	[252]

## Data Availability

The data presented in this study are available in this article.

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
