# Peer review of "Beyond NPK: Mineral Nutrient-Mediated Modulation in Orchestrating Flowering Time"

_plants, 2023, doi:10.3390/plants12183299_

Round 1

Reviewer 1 Report

Review report:

  • A brief summary 

The present review explored transcription factors and enzymes that rely on micronutrients as cofactors for their functional activation or catalytic activity enhancement of flowering time and enhanced understanding of nutrient-mediated floral induction and floral regulatory mechanisms.

  • General concept comments

Overall, this review presents valuable information on how secondary nutrients (calcium, magnesium, and sulfur) and micronutrients (boron, zinc, iron, and copper) influence the flowering process in plants. In addition, enzymes and cofactors involved in the nutrient element-mediated regulation of the switch to flower or flowering time in various plants are comprehensively presented. The review provides an advancement of the current knowledge and fits the journal scope. The tables show the data properly and easily interpreted. Most cited references are recent (within the last 5 years). Otherwise, the manuscript still has some problems in conclusions which should be drawn and supported more coherently. Additionally, authors should clarify which nutrients have dealt with in the review and use proper terminology. For this, I recommend it to be considered after some Minor revision.

Specific comments

·         Title: The title ‘Mineral nutrient-mediated modulation of flowering time’ needs to be reworded because it is too general and does not describe the impact of nutrients in the biochemical processes of flowering such as enzymes and cofactors analyzed in the review.

·         Line 14-19: Nitrogen, phosphorus and potassium are macronutrients that can be further defined as primary nutrients. Calcium, magnesium and sulfur are macronutrients which can be further defined as secondary nutrients, while boron, zinc, iron, and copper are micronutrients. Please reword correctly.

·         Line 627-648: The review article doesn’t refer only to micronutrients but also to macronutrients. Please define correctly which nutrients you have dealt with in the review.

·         Line 646-648: Please support your conclusions and define how this review can enhance research for crop breeding.

Minor editing of English language required

Author Response

Attached is a file with the authors' responses to reviewers' comments, and the revised manuscript. 

Thank you. 

Reviewer 2 Report

1- Increase number of keywords to 5 keywords and write the keywords on the basis of alphabetic order. Please, try to use keywords which are not in the title of the manuscript.

2- In the referencing in the text, please, pay attention to format and instruction of the journal. For example, if it is (1,2,3,4,5,6), you should write it as (1-6); or if it is (1,3,4,5,6,7), like line 40, you should write it as (1,3-7).

3- If you are review is Systematic Review, open one section as Materials and Methods and then write one paragraph about the methods which have you used in your manuscript and mention from which sources you have gathered information like PubMed, Scopus, etc.

4- I think the font of Table 1 is not on the basis of journal s format. Please, double check with Instruction for authors.

5- Why you references are not on the basis of orders in table 1??? from 40, it has moved to 85. Referencing in the text should be done on the basis of number orders.

6- Line 79, it should be written as follow (25-29), not (25,26,27,28,29)

7- Section 4, it should be Conclusion and Perspective, and if in the plan of authors these two sections are different from each others, write one section as Perspective, and one Section as Conclusion.

8- Reference 68 and Reference 101, write the number of full authors, do not use et al. in any References in your manuscript.

Author Response

(The authors gave the same response as above.)
